# Provider-initiated HIV testing and TB screening in the era of universal coverage: Are the right people being reached? A cohort study in Blantyre, Malawi

Luke Mair[1,2]*, Elizabeth L. Corbett[3,4], Helena R. A. Feasey[3,4], Wala Kamchedzera[4], McEwen Khundi[3,4], David G. Lalloo[1], Hendramoorthy Maheswaran[5], Marriott Nliwasa[6], S. Bertel Squire[1], Emily L. Webb[7], Peter MacPherson[1,4]

1 Department of Clinical Sciences, Liverpool School of Tropical Medicine, Liverpool, United Kingom, 2 Tropical and Infectious Diseases Unit, Royal Liverpool University and Broadgreen Hospitals NHS Trust, Liverpool, United Kingdom, 3 Department of Clinical Research, London School of Hygiene and Tropical Medicine, London, United Kingdom, 4 Malawi-Liverpool-Wellcome Trust Clinical Research Programme, Blantyre, Malawi, 5 Department of Public Health and Policy, University of Liverpool, Liverpool, United Kingdom, 6 Helse-Nord TB Programme, College of Medicine, University of Malawi, Malawi, 7 MRC Tropical Epidemiology Group, London School of Hygiene and Tropical Medicine, London, United Kingdom

* luke.mair1@nhs.net

## Abstract

### Introduction

Patients with tuberculosis (TB) symptoms have high prevalence of HIV, and should be prioritised for HIV testing.

### Methods

In a prospective cohort study in Bangwe primary care clinic, Blantyre, Malawi, all adults (18 years or older) presenting with an acute illness were screened for TB symptoms (cough, fever, night sweats, weight loss). Demographic characteristics were linked to exit interview by fingerprint bioidentification. Multivariable logistic regression models were constructed to estimate the proportion completing same-visit HIV testing, comparing between those with and without TB symptoms.

### Results

There were 5427 adult attendees between 21/5/2018 and 6/9/2018. Exit interviews were performed for 2402 (44%). 276 patients were excluded from the analysis, being already on antiretroviral therapy (ART). Presentation with any TB symptom was common for men (54.6%) and women (57.4%). Overall 27.6% (585/ 2121) attenders reported being offered testing and 21.5% (455/2121) completed provider-initiated HIV testing and counselling (PITC) and received results. The proportions offered testing were similar among participants with and without TB symptoms (any TB symptom: 29.0% vs. 25.7%). This was consistent for each individual symptom; cough, weight loss, fever and night sweats. Multivariable

**Data Availability Statement:** Data and code to reproduce these analyses are available as an R

package from GitHub (https://github.com/petermacp/hiv.prospect).

**Funding:** Peter MacPherson is funded by Wellcome (206575). Elizabeth L Corbett is funded by Wellcome (200901). Emily L Webb is funded by the UK Medical Research Council (grant reference MR/R010161/1), this award is jointly funded by the UK Medical Research Council (MRC) and the UK Department for International Development (DFID) under the MRC/DFID Concordat agreement and is also part of the EDCTP2 programme supported by the European Union. The funders were not involved in study design, data collection, analysis, decision to publish or preparation of the manuscript.

**Competing interests:** The authors have declared that no competing interests exist.

regression models indicated men, younger adults and participants who had previously tested were more likely to complete PITC than women, older adults and those who had never previously tested.

## Conclusions

Same-visit completion of HIV testing was suboptimal, especially among groups known to have high prevalence of undiagnosed HIV. As countries approach universal coverage of ART, identifying and prioritising currently underserved groups for HIV testing will be essential.

## Introduction

The deadline of the Joint United Nations Programme 90-90-90 campaign to end the HIV epidemic by 2020 is approaching.[1] Substantial global progress has been made with seven countries having reached the target of achieving high coverage of viral suppression amongst all people living with HIV.[2] Malawi is among the large proportion of countries still striving to reach these targets.

Through initiatives such as provider-initiated testing and counselling (PITC), and with the strong support of the Malawi National HIV Programme, 70% of Malawians knew their status in 2016.[2] Despite this being an improvement on previous years, there remains substantial room for further improvement.

Current Malawian National Guidelines focus on facility testing and community testing as priority opportunities for HIV testing and diagnosis.[3] PITC is recommended for anyone attending healthcare facilities including outpatient departments as one of the most effective strategies for identifying new HIV diagnoses. Other strategies recommended include index testing and self-testing. [3] Community-based HIV testing strategies have helped to reach the "untested" population, including key populations with high HIV prevalence that are hard to reach, such as men who have sex with men, female sex workers, people who inject drugs, and transgender women.[4] Despite impressive achievements in increasing community-based strategies, the majority of HIV testing still occurs in health facilities. PITC in health facilities remains a critical strategy for identifying and linking to care people with advanced HIV and concomitant high risk of death. When investigated previously in Blantyre, Malawi in 2011 only 13% of all adult clinic attenders completed PITC, leading to many missed opportunities for diagnosis and linkage to ART.[5]

New strategies for improving rates of HIV test offer within health facilities are essential to help reach the untested population. One strategy is routine opt-out (ROOT) HIV testing in which patients are offered an HIV test at each clinic visit unless they specifically refuse.[6] A further is utilising semi-supervised oral fluid based HIV testing kits within facilities.[7]

Patients presenting with symptoms of tuberculosis (TB)–cough (of any duration), fever, weight loss and night sweats–have a high prevalence of undiagnosed HIV and advanced immunosuppression.[8] In fact, in settings with generalised HIV epidemics those presenting with TB symptoms have higher prevalence of HIV than people with confirmed TB.[9] Despite the high HIV prevalence in patients with TB symptoms, they are not listed as a priority screening group in the Malawian National Strategic Plan for HIV and AIDS.[10]

This study investigated PITC uptake for those with and without TB symptoms at a busy primary health care centre in Malawi. The objectives were to describe the fraction of acute adult

clinic attenders who were offered and received HIV testing and counselling, and to explore risk factors associated with non-completion of PITC.

## Methods

### Study design and participants

We undertook a prospective study among adults (18 years or older) attending Bangwe Health Centre (BHC) with an acute illness between 21st May 2018 and 6th September 2018. BHC is a busy public primary health care clinic located in urban Blantyre, Malawi, where adult HIV prevalence is 18%[11] and TB prevalence approximately 1%.[12] This study was nested within the pilot phase of a pragmatic randomised trial being conducted at the clinic that aims to evaluate strategies to optimise diagnosis and treatment initiation for HIV and tuberculosis. [13]

BHC has an adult outpatient department where community members can self-present and are registered and receive care free-of-charge for illnesses. Within BHC, there are additionally HIV, TB, antenatal, and paediatrics departments from where we did not recruit study participants. At BHC there are no physicians; medical care is provided by nurses and clinical officers. PITC is recommended by the Malawi National HIV Programme for all attendances to an outpatient department, and was available at BHC, provided by Ministry of Health-accredited counsellors. ART and antituberculosis therapy were provided onsite at the HIV care clinic and TB clinic respectively. Sputum smear microscopy and GeneXpert MTB/Rif testing were available onsite in the clinic laboratory. Patients requiring chest X-ray for investigation of possible TB required healthcare referral to the city's central hospital.

### Screening and exit interviews

Research assistants stationed at the clinic's outpatient department registration desk identified all adult individuals attending the clinic to receive acute care. Following verbal consent, a fingerprint scan was recorded for bioidentification. Research assistants then undertook a brief screen, recording clinic attenders' age, sex and the presence of TB symptoms (cough and cough duration, fever, weight loss, and night sweats). Information was not recorded from participants who declined fingerprint scanning. Following the completion of screening, clinic attenders joined the outpatient department queue to be seen by health workers.

Next, participants were reviewed by a health worker. Health workers worked independently from research assistants and were unaware of the results of questions asked during initial screening. Health workers were not required to screen for TB symptoms as part of study protocol, however it is recommended in national and WHO guidelines that all clinic attenders be: 1) Screened for TB symptoms and 2) Offered PITC.[14]

Research assistants were also based at the two clinic outpatient department exits, where they identified adults leaving the department and invited them to participate in an exit interview. The layout of the clinic is such that all participants exiting the clinic would pass the research assistants. Though researchers tried to select as many patients as possible, due to limitations in the number of research assistants not all participants completed exit interview. To minimise inconvenience and maximise research assistants' capacity to complete interviews, questionnaires were brief with predominantly binary responses to questions rather than in-depth qualitative questions. Prior to exit interview identity was verified by fingerprint bioidentification. Clinic attenders were asked about the care they received whilst in the clinic, including whether they had been offered HIV testing, and whether HIV testing had been completed during their clinic visit. We additionally recorded whether participants had ever previously tested for HIV, their self-reported HIV and ART status, and whether they had previously been treated for TB.

## Statistical methods

We summarised characteristics of clinic attenders using proportions, medians and means, and compared them between those who did and did not undergo clinic exit interview. Individuals who reported cough of any duration, fever, weight loss, or night sweats were classified as having "any TB symptom". Cough duration was also recorded and in accordance with WHO guidelines, we defined chronic cough as self-reported cough lasting greater than two weeks.

Participants with known HIV-positive status who were already taking ART were excluded from the analysis. Those with self-reported HIV positive status who were not taking ART were included as this provides further opportunity for confirmatory testing and linkage to ART. We summarised the proportion of clinic attenders who reported having been offered an HIV test and who reported having completed an HIV test on the same day as their visit, and compared groups by demographic and clinical characteristics. As previous research suggests men have a higher burden of undiagnosed TB and present later for diagnoses than women, results were stratified by sex to investigate differences between sexes. [8]

To investigate characteristics associated with reporting completing HIV testing during a clinic visit exit interview, we constructed Bayesian multivariable logistic regression models. [15] Initial models included terms for age and sex, and subsequent models evaluated the incremental improvement in model fit from adding reported history of previous TB treatment and having previously tested for HIV. We then constructed models including terms for the presence of either: cough; chronic cough; or any TB symptoms. Weakly regularising priors were applied to model intercepts and slopes. Convergence was evaluated by inspecting trace plots and effective sample numbers, and by calculating Gelman-Rubin statistics. Model fits were compared using the Watanabe-Akaike information criteria (WAIC) statistic and weights.

Three thousand samples were drawn from each model posterior distribution using Markov Chain Monte Carlo methods implemented within Stan. Model coefficients on the log-odds scale were exponentiated to give odds ratios, and posterior means and 89% probability intervals were calculated and plotted. Posterior predictions for age and sex were graphed, stratified by presence of TB symptoms (cough, chronic cough, any TB symptom). Analysis was conducted using R version 3.5.2 (R Foundation for Statistical Computing, Vienna).

## Ethical approval

Ethical approval was received from the College of Malawi Research Ethics Committee (COMREC), and from the Liverpool School of Tropical Medicine Research Ethics Committee. Individuals undergoing exit interviews provided written (or witnessed thumb-print if illiterate) consent.

## Data and reproducibility

Data and code to reproduce these analyses are available as an R package from https://github.com/petermacp/hiv.prospect.

## Results

### Characteristics of clinic attenders

Between 21st May and 6th September 2018, a total of 5427 adult acute attendances were recorded at the study clinic. Five individuals declined fingerprint identification and were not included in the study. Of the remaining 5422 attendances, 4895 (90%) were by individuals who attended the clinic only once during the study period, and 527 (10%) by individuals who attended more than once; the maximum number of clinic attendances during the study period

was seven, recorded by one individual. The number of daily acute clinic attendances recorded ranged between 41 and 106 (mean = 70.4, median = 69).

Overall, 64% (3486/5422) of clinic attendance episodes were by women, with similar age distributions between women (median: 28 years, range: 18–88) and men (median: 28 years, range: 18–89)–Table 1. Women and men attending the clinic had similar distributions of reported: cough of any duration (women: 1135, 33%; men: 686, 35%); chronic cough (women: 218, 11.3%; men: 304, 8.7%); weight loss (women: 431, 12.4%; men: 281, 14.5%); fever (women: 951, 27.3%; men: 630, 32.5%); night sweats (women: 596, 17.1%; men: 388: 20.0%); and any TB symptom (women: 1905, 54.6%; men: 1111, 57.4%). Overall, 1.4% (77/5422) of clinic attenders reported being currently treated for TB (women: 48, 1.4%; men: 29, 1.5%).

A total of 2397 (44%) clinic attendances were followed by same-day study exit interviews. Characteristics between clinic attenders who did and did not complete exit interviews were similar, but with some differences (Table 1). Women (1497, 42.9%) were less likely to complete exit interviews than men (900, 46.5%). Additionally, clinic attenders who reported any TB symptom were more likely to complete an exit interview compared to clinic attenders who didn't (45.4% vs 42.7%). This was consistent for cough, fever, weight loss and night sweats. However those with chronic cough were less likely to complete exit interview than those without (42.3% vs 44.4%).A total of 276 (11.5%) clinic attenders reported that they were HIV positive and taking ART and were exclude from regression modelling.

**Table 1. Characteristics of adult acute clinic attendances.**

| | Exit Interviewed (N = 2397) | Not Exit Interviewed (N = 3025) | Total (N = 5422) |
|---|---|---|---|
| Sex | | | |
| Female | 1497 (42.9%) | 1989 (57.1%) | 3486 (100.0%) |
| Male | 900 (46.5%) | 1036 (53.5%) | 1936 (100.0%) |
| Age | | | |
| Median (Range) | 28 (18, 89) | 27 (18, 89) | 28 (18, 89) |
| Cough | | | |
| No | 1548 (43.0%) | 2053 (57.0%) | 3601 (100.0%) |
| Yes | 849 (46.6%) | 972 (53.4%) | 1821 (100.0%) |
| Weight Loss | | | |
| No | 2081 (44.2%) | 2629 (55.8%) | 4710 (100.0%) |
| Yes | 316 (44.4%) | 396 (55.6%) | 712 (100.0%) |
| Fever | | | |
| No | 1663 (43.3%) | 2178 (56.7%) | 3841 (100.0%) |
| Yes | 734 (46.4%) | 847 (53.6%) | 1581 (100.0%) |
| Night Sweats | | | |
| No | 1936 (43.6%) | 2502 (56.4%) | 4438 (100.0%) |
| Yes | 461 (46.8%) | 523 (53.2%) | 984 (100.0%) |
| Any TB Symptom[†] | | | |
| No | 1027 (42.7%) | 1379 (57.3%) | 2406 (100.0%) |
| Yes | 1370 (45.4%) | 1646 (54.6%) | 3016 (100.0%) |
| Chronic Cough[¶] | | | |
| No | 2176 (44.4%) | 2724 (55.6%) | 4900 (100.0%) |
| Yes | 221 (42.3%) | 301 (57.7%) | 522 (100.0%) |

† Any TB symptom: cough, or weight loss, or fever, or weight loss.

¶ Cough of 14 days or longer

Of 2121 patients included in the analysis 1907 (89.9%) self-reported previously ever testing for HIV, in comparison to 214 (10.1%) who self-reported never testing. Females (1194/1295, 92.2%) were more likely to self-report previous HIV testing than males (713/826, 86.3%). Previous testers (median: 27 years, range 18–87), were of similar age to non-testers (mean: 26 years, range 18–89).

## Offer and completion of HIV testing

A total of 585/2121 (27.6%) eligible participants reported being offered HIV testing by a health worker during their clinic attendance, and 455/2121 (21.5%) completed HIV testing. Twenty-nine percent of men (242/826) and twenty-six percent (343/1295) of women were offered HIV testing by health workers. Those offered HIV testing were younger than those not (median age: 26 vs. 29 years respectively). There were few differences in the proportions offered HIV testing among participants who reported and did not report TB symptoms (any TB symptom: 348/1199, 29.0% vs. 237/922, 25.7%; cough: 211/732, 28.8% vs. 374/1389, 26.9%; chronic cough: 39/179, 21.7% vs. 546/1942, 28.1%; weight loss: 74/252, 29.3% vs. 511/1869, 27.3%; fever: 190/647, 29.3% vs. 395/1474, 26.8%; night sweats: 104/407, 25.6% vs. 481/1714, 28.1%). Participants who reported previously having been treated for TB were less likely to be offered HIV testing (16/78, 20.5%) than those not previously treated for TB (569/2043, 27.9%). Participants who self-reported previously testing for HIV were more likely to be offered testing (539/1907, 28.3%), than those who had not previously HIV tested (46/214, 21.5%).

In addition to being more likely to be offered HIV testing, men (184/242, 76.0%) were more likely to report completing HIV testing compared to women (252/343, 73.4%). Age was also associated with completion of HIV testing, with the median age of those completing testing (26 years) lower than those who didn't complete HIV testing (29 years). Similarly, patterns of completion of HIV testing among those with and without TB symptoms closely mapped data for offer of HIV testing (cough: 165/211, 78.2% vs. 271/374, 72.5%; chronic cough: 28/39, 71.8% vs. 408/546, 74.7%; weight loss: 54/74, 73.0% vs. 382/511, 74.8%; fever: 135/190, 71.1% vs. 301/395, 76.2%; night sweats: 74/104, 71.2%% vs. 362/481, 75.3%; any TB symptom: 254/348, 73.0% vs. 182/237, 76.8%). Participants who had previously been treated for TB (9/16, 56.25%) were less likely to complete HIV testing than those hadn't previously been treated (427/659, 75.0%). Participants who self-reported previous HIV testing were likelier to completing testing (417/539, 77.4%) than those who had not (19/46, 41.3%).—(Fig 1).

In the initial regression model (Table 2, Model 1), men were more likely to complete HIV testing than women (posterior odds ratio: 1.15, 89% interval: 0.97–1.37), and older participants were less likely to complete HIV testing (posterior odds ratio per year increase in age: 0.99, 89% interval: 0.98–0.99). Models that included terms for cough (Models 2a, 2b and 2c) additionally showed that reporting cough, and having previously tested for HIV increased the posterior odds of HIV testing, whilst previously being treated for TB decreased the posterior odds of completing HIV testing. In contrast, having chronic cough instead of cough (Models 3a, 3b, 3c) resulted in a reduced posterior odds ratio of completing HIV testing. Reporting any TB symptom (Models 4a, 4b and 4c) was linked to a reduced posterior odds ratio for completing HIV testing. Models that included terms for age, sex, previous HIV testing, previous TB treatment, and either cough, chronic cough or any TB symptom (Models 2c, 3c and 4c) had the lowest WAIC statistics and greatest weight. Model summary estimates are provided in Table 2 and visually in Fig 2.

When converted to the probability scale, there was strong evidence of decreased probability of completing HIV testing with increasing age across all three models that included terms for TB symptoms (Models 2c, 3c and 4c). Additionally, the posterior probability of HIV testing

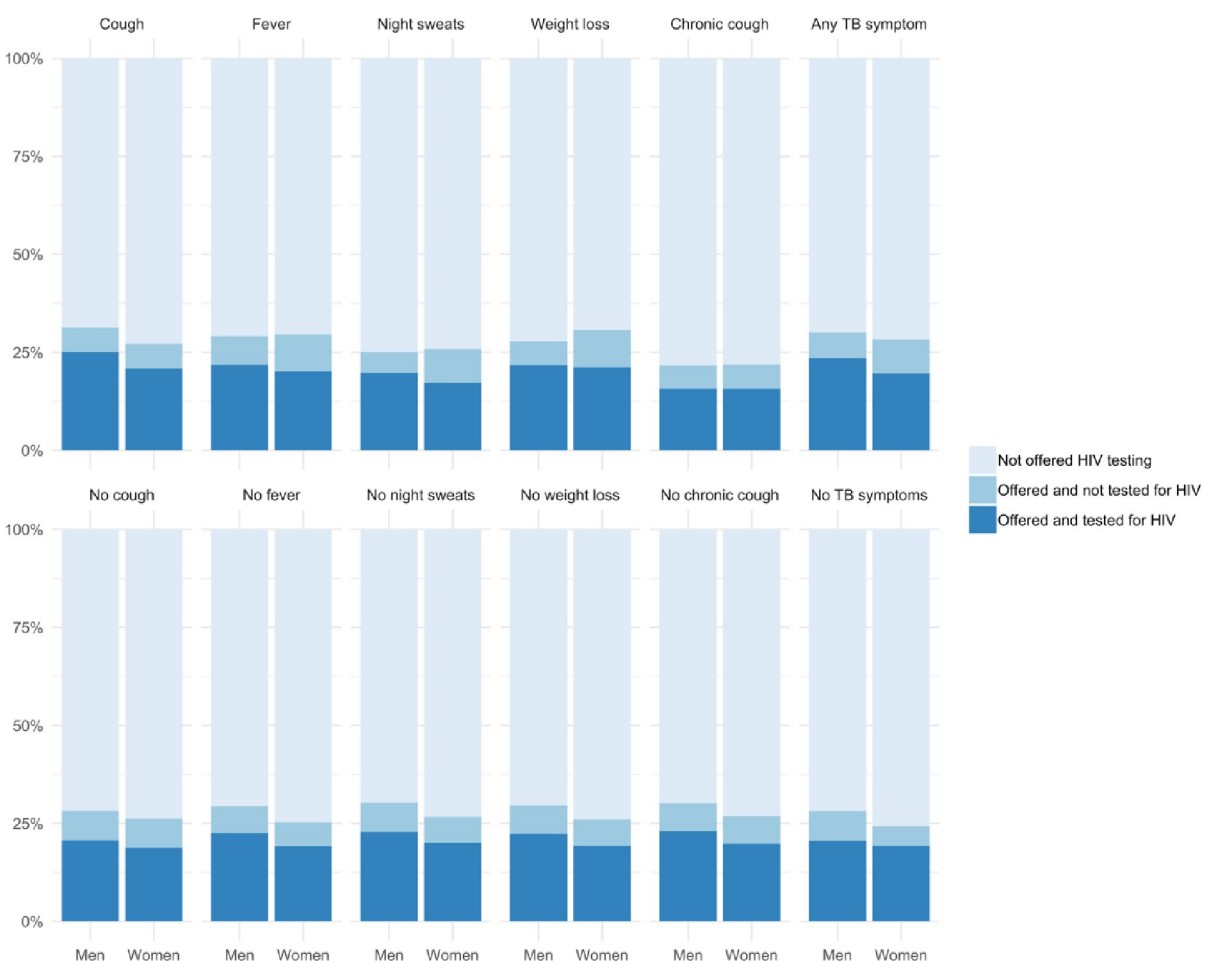

**Fig 1. Percentages of participants offered and completing HIV testing by TB symptoms.** Proportions of patients offered HIV testing. Stratified by those with TB symptoms (top row) and those without TB symptoms (bottom row).

was consistently higher among men than women, as well as those with a prior self-reported history of HIV testing. (Fig 3).

## Discussion

The findings of our study reveal that less than one-third (27.6%) of adults attending a primary health care clinic with an acute care episode were offered an HIV test, and only 21.4% completed same-day HIV testing. Despite this being an improvement on when previously examined in Blantyre in 2010 (13%), [5] this study reveals many missed opportunities for HIV testing and diagnosis. This highlights lack of implementation of national guidelines, which state that all patients attending an outpatient appointment should be offered an HIV test.

A recent overview of delivery of facility-based HIV testing services in Malawi showed that the majority of health centres use symptom-based PITC, with more unwell people prioritised for HIV testing.[16] Understanding the rationale for this and developing new strategies are key for further improvement. Previous qualitative research has revealed that health care workers felt more compelled to provide PITC in ANC than in outpatients departments.[17] Health-care workers also saw ART as a precious resource which should be reserved for those who were more unwell in outpatient departments. Utilising symptom-based PITC leads to

**Table 2. Model posterior probabilities of completing HIV testing during clinic visit.**

| Parameter | Posterior probability (mean odds ratio, 89% interval) | | | | | | | | | |
|---|---|---|---|---|---|---|---|---|---|---|
| | Model 1 | Model 2a | Model 2b | Model 2c | Model 3a | Model 3b | Model 3c | Model 4a | Model 4b | Model 4c |
| Intercept | 0.35 (0.25–0.49) | 0.28 (0.19–0.42) | 0.51 (0.26–1.00) | 0.06 (0.02–0.16) | 0.35 (0.25–0.50) | 0.62 (0.33–1.18) | 0.07 (0.03–0.20) | 0.42 (0.28–0.62) | 0.76 (0.39–1.55) | 0.09 (0.03–0.24) |
| Male sex | 1.15 (0.97–1.37) | 1.15 (0.96–1.36) | 1.16 (0.97–1.39) | 1.23 (1.01–1.42) | 1.16 (0.98–1.39) | 1.17 (0.99–1.39) | 1.24 (1.04–1.49) | 1.15 (0.97–1.36) | 1.17 (0.99–1.38) | 1.23 (1.05–1.46) |
| Age (Per Year) | 0.99 (0.98–0.99) | 0.98 (0.98–0.99) | 0.99 (0.98–0.99) | 0.99 (0.98–0.99) | 0.99 (0.98–0.99) | 0.99 (0.98–0.99) | 0.99 (0.98–0.99) | 0.98 (0.98–0.99) | 0.99 (0.98–0.99) | 0.99 (0.98–0.99) |
| Previous TB | | | 0.53 (0.29–0.91) | 0.50 (0.28–0.84) | | 0.56 (0.31–0.99) | 0.52 (0.28–0.90) | | 0.54 (0.31–0.91) | 0.50 (0.28–0.86) |
| Ever Tested For HIV | | | | 3.02 (2.10–4.43) | | | 2.96 (2.05–4.46) | | | 3.02 (2.07–4.41) |
| Cough | | 1.21 (1.02–1.44) | 1.22 (1.03–1.46) | 1.24 (1.04–1.47) | | | | | | |
| Chronic Cough | | | | | 0.82 (0.59–1.14) | 0.84 (0.60–1.17) | 0.84 (0.58–1.18) | | | |
| Any TB Symptom | | | | | | | | 0.90 (0.76–1.06) | 0.90 (0.75–1.07) | 0.88 (0.74–1.05) |
| WAIC* | 2196.0 | 2194.9 | 2193.6 | 2169.7 | 2197.3 | 2196.7 | 2173.2 | 2196.9 | 2195.7 | 2171.9 |
| Weight | 0.00 | 0.00 | 0.00 | 0.66 | 0.00 | 0.00 | 0.12 | 0.00 | 0.00 | 0.22 |

*WAIC = Watanabe-Akaike information criteria

detection of patients later in their disease course with greater risk of suboptimal HIV treatment outcomes. Moreover, certain groups who are difficult to reach through existing programmes and who have higher rates of undiagnosed HIV (such as older people) are disadvantaged.

Changes in approach are needed to reduce further missed opportunities for testing. One strategy would be to include a ROOT approach in national guidelines whereby patients are offered an HIV test for every acute care attendance unless they decline, potentially leading to higher uptake of HIV testing. [6] Using the ROOT approach is resource intensive and common barriers to reaching 100% coverage include limited health worker capacity, inadequate space in facilities, and insufficient test kits.[5,16] These stated limitations may account for the population of patients in our study who were offered HIV testing but did not undergo a test.

A second approach would be utilising semi-supervised oral fluid-based rapid HIV self-testing kits within facilities to help overcome some of these barriers. This has previously been implemented with success in a rural African setting with high rates of accuracy and user acceptance.[7] Qualitative research has suggested that patients who noticed a decline in their health, may wish to know their status and therefore may be motivated to test themselves.[17] This strategy may be key to removing the heavy burden of work on the healthcare workers and empowering patients to know their own status.

In generalised HIV epidemics, people with symptoms of TB have a high prevalence of HIV at all levels of healthcare; including community level, primary health care and hospital inpatients.[9] In acknowledgement of these patients being a high risk group, WHO policy on collaborative TB/HIV activities was updated in 2012 to advise that in generalised HIV epidemics routine HIV testing should be offered to all with presumptive or diagnosed TB.[18]

Contrary to our expectation of higher HIV testing completion among people with TB symptoms, we found that the presence of TB symptoms–defined by either cough of any duration, chronic cough, or any TB symptom–was not associated with HIV testing; in fact having any TB symptom reduced posterior odds of completing HIV testing. Considering over 50% of

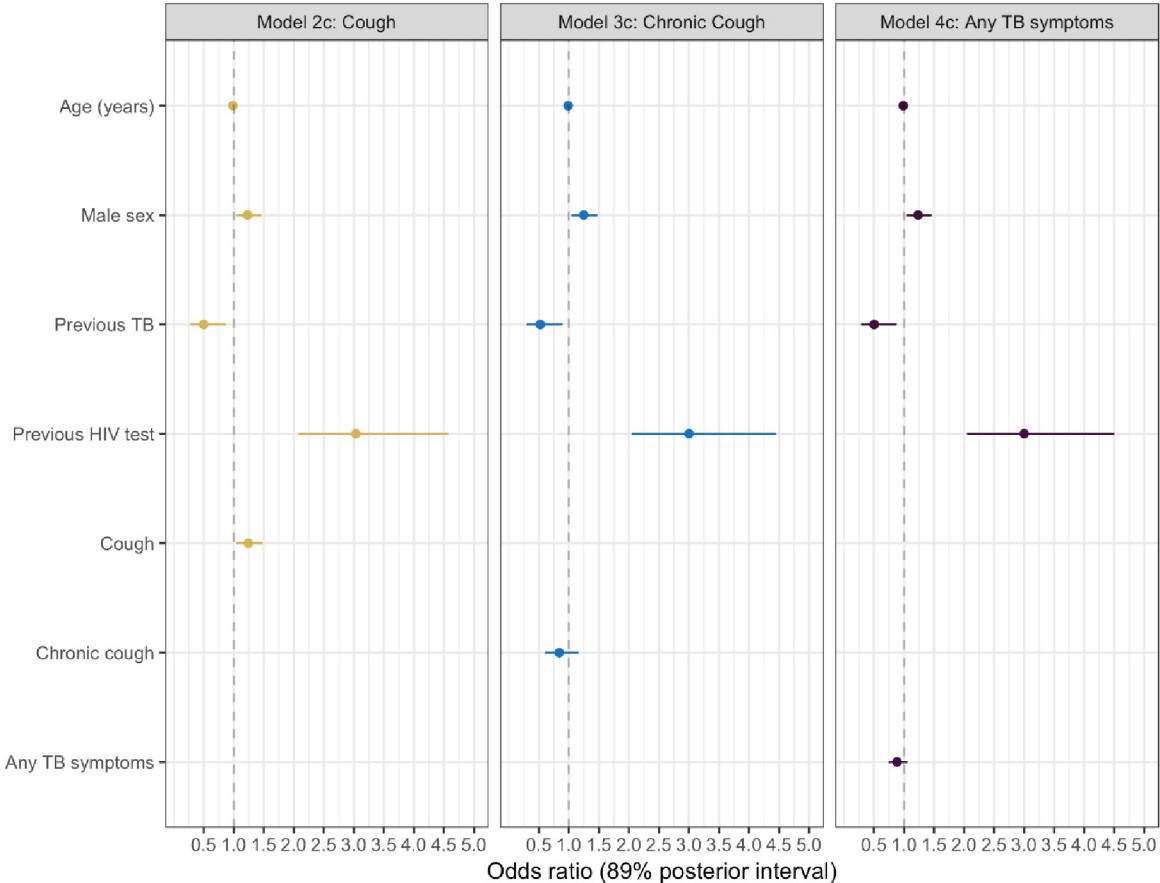

**Fig 2. Model posterior probabilities for completing HIV testing during clinic attendance for individuals with and without TB symptoms.**

clinic attenders in our study reported symptoms of TB, our findings suggest large numbers of missed opportunities for new diagnoses of HIV and linkage to ART.

The Malawian Ministry of Health National Tuberculosis Control Programme recommends PITC HIV testing for all patients with presumptive TB.[19] This issue is also recognised in the Malawian National Strategic Plan for HIV AIDS 2015–2020 which suggests PITC HIV testing should be moved upstream to include patients with suspected TB as well as confirmed TB; however this is not listed as a priority intervention.[10] The findings in this study combined with review of the literature suggest patients with TB symptoms should be list as a priority intervention for expanding PITC, and further collaboration is needed between TB and HIV coordinators at a national level.

With the maturation of the HIV epidemic in Southern Africa, the prevalence of HIV (and undiagnosed HIV) is growing among older adults.[20] This key population, who often remain sexually active, are challenging to reach with community-based HIV testing interventions.[21] Our study shows facility rates of offering and acceptance of HIV testing decreased with increasing age, again suggesting missed opportunities for HIV diagnoses. Efforts should be made to incorporate HIV testing into guidelines for those with non-communicable diseases, which are likely to be "unmasked" by this aging population. Men were more likely to test than women which could be attributed to presenting later in their disease with more advanced clinical symptoms and therefore triggering symptom-based PITC. Only 41.3% of patients who self-

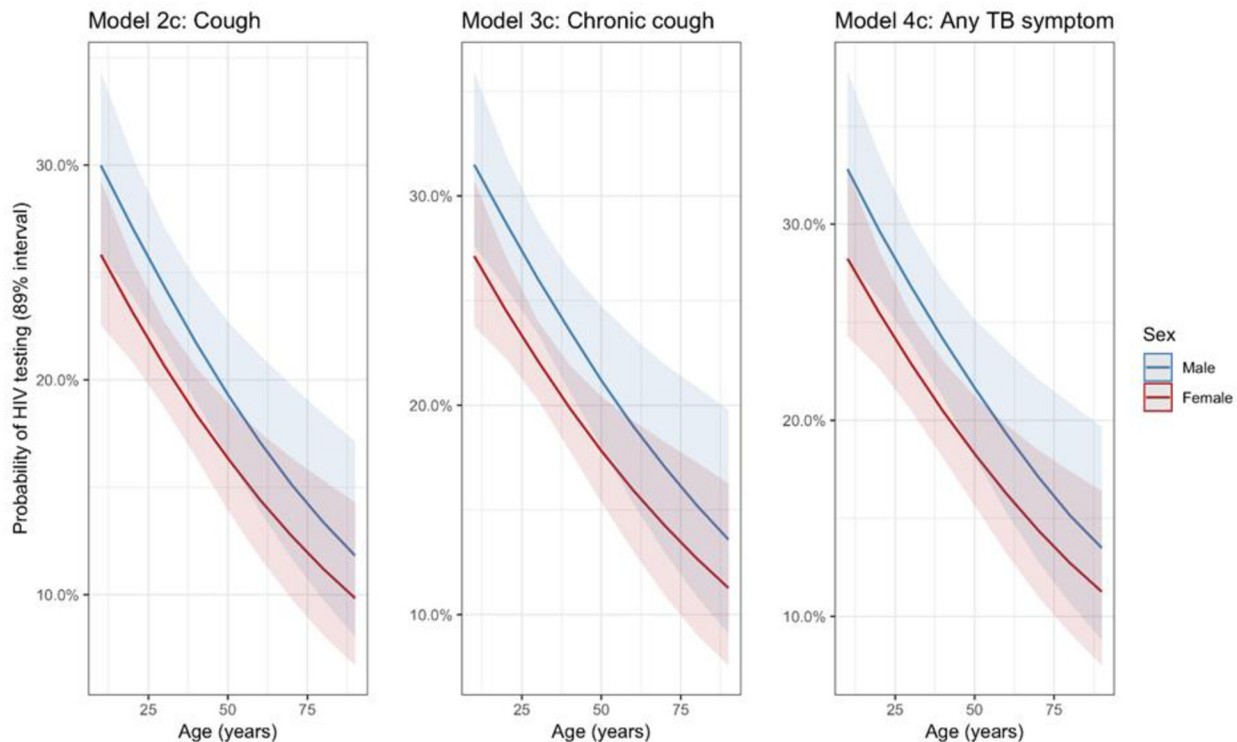

**Fig 3. Predicted probability of HIV testing by age and sex.** Predicted probabilities are estimated from multivariable Bayesian regression models, adjusted for no history of previous TB and not having tested previously for HIV.

reported having never previously tested were offered PITC. Utilising ROOT can help prevent missing vital opportunities for this population to learn their status.

Participants with a self-reported history of HIV testing have previously undertaken pre and post-test counselling and have overcome barriers of stigma relating to testing. This awareness of the importance of knowing their status may drive them to request further HIV testing from health-workers. This population of frequent HIV testers may also be undertaking higher risk activities or presenting with conditions that demand frequent testing such as sexually transmitted infections.

Our study does have limitations. Firstly due to limited availability of Research Assistants, exit interviews were only performed in 44% of visits potentially resulting in bias in selection of participants. It is possible that participants who were sicker were being referred to same day secondary facilities and were under greater time-pressure and less likely to be stopped or indeed engage with exit interview. Secondly symptoms and HIV testing patterns were self-reported which could result in bias in measurement of outcomes. Thirdly patients were not asked the time since last HIV test in exit interview. Therefore, an analysis of proportion of patients testing annually in accordance with current WHO guidelines could not be performed. Finally, results were taken from one health facility and may not be representative of findings throughout the country and continent.

## Conclusion

Same-visit completion of PITC in primary care was poorly implemented, especially among groups known to have high prevalence of undiagnosed HIV (people with TB symptoms, older adults). As countries approach universal coverage of ART, identifying and prioritising

currently underserved groups for HIV testing will be essential. Routine opt-out HIV testing needs to be prioritised if HIV programmes are to identify and treat the missing millions with undiagnosed HIV.

## Supporting information

**S1 Questionnaire.**
(XLSX)

## Author Contributions

**Conceptualization:** Luke Mair, Elizabeth L. Corbett, Helena R. A. Feasey, Wala Kamchedzera, McEwen Khundi, David G. Lalloo, Hendramoorthy Maheswaran, Marriott Nliwasa, S. Bertel Squire, Emily L. Webb, Peter MacPherson.

**Data curation:** Peter MacPherson.

**Formal analysis:** Luke Mair, Peter MacPherson.

**Funding acquisition:** Peter MacPherson.

**Investigation:** Luke Mair, Peter MacPherson.

**Methodology:** Peter MacPherson.

**Project administration:** Peter MacPherson.

**Resources:** Peter MacPherson.

**Supervision:** Peter MacPherson.

**Visualization:** Peter MacPherson.

**Writing – original draft:** Luke Mair, Peter MacPherson.

**Writing – review & editing:** Luke Mair, Elizabeth L. Corbett, Helena R. A. Feasey, Wala Kamchedzera, McEwen Khundi, David G. Lalloo, Hendramoorthy Maheswaran, Marriott Nliwasa, S. Bertel Squire, Emily L. Webb, Peter MacPherson.

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
