## [Decision Letter · Decision Letter 0]

19 Feb 2020

PONE-D-19-35399

Provider-initiated HIV testing and TB screening in the era of universal coverage: are the right people being reached? A cohort study in Blantyre, Malawi

PLOS ONE

Dear Dr. Mair,

Thank you for submitting your manuscript to PLOS ONE. After careful consideration, we feel that it has merit but does not fully meet PLOS ONE’s publication criteria as it currently stands. Therefore, we invite you to submit a revised version of the manuscript that addresses the points raised during the review process. Please review the feedback and requests from clarification from the two reviewers, and provide a detailed point-by-point response.

We would appreciate receiving your revised manuscript by Apr 04 2020 11:59PM. To enhance the reproducibility of your results, we recommend that if applicable you deposit your laboratory protocols in protocols.io, where a protocol can be assigned its own identifier (DOI) such that it can be cited independently in the future. For instructions see: http://journals.plos.org/plosone/s/submission-guidelines#loc-laboratory-protocols

We look forward to receiving your revised manuscript.

Kind regards,

Susan Marie Graham, MD, MPH, PhD

Academic Editor

PLOS ONE

Journal Requirements:

1. Please include additional information regarding the survey or questionnaire used in the study and ensure that you have provided sufficient details that others could replicate the analyses. For instance, if you developed a questionnaire as part of this study and it is not under a copyright more restrictive than CC-BY, please include a copy, in both the original language and English, as Supporting Information.

Reviewers' comments:

Reviewer's Responses to Questions

**Comments to the Author**

1. Is the manuscript technically sound, and do the data support the conclusions?

Reviewer #1: Partly

Reviewer #2: Yes

2. Has the statistical analysis been performed appropriately and rigorously? 

Reviewer #1: Yes

Reviewer #2: Yes

3. Have the authors made all data underlying the findings in their manuscript fully available?

Reviewer #1: Yes

Reviewer #2: Yes

4. Is the manuscript presented in an intelligible fashion and written in standard English?

Reviewer #1: Yes

Reviewer #2: Yes

5. Review Comments to the Author

Reviewer #1: Overall

This manuscript describes a prospective cohort study undertaken in an adult primary care clinic in Blantyre Malawi to evaluate whether clinic attendees were offered PITC and to compare same day HIV testing rates between participants reporting TB symptoms vs those reporting no TB symptoms. From other data, rates of undiagnosed HIV are high among clinic clients reporting TB symptoms, therefore should be prioritized for HIV testing. The investigators relied on self-report of HIV-testing upon exit interview, and only 44% of clinic attendees underwent exit interview.

The main finding of the study was that among participants undergoing exit interview, only 27.6% reported being offered testing, with 21.5% completing PITC and receiving HIV test results. Testing offered was relatively similar among those with and without TB symptoms. The multivariable regression model indicated men, younger adults, and participants who had previously undergone HIV-testing were more likely to complete HIV testing than women, older adults, and those who had not previously been tested for HIV.

Although an important topic- universal HIV testing as well as importance of HIV testing among those with TB symptoms, the paper could be strengthened by a number of additions/clarification:

-The authors do not included data regarding proportion of participants who report TB symptoms (and those without TB symptoms) who were diagnosed with HIV. This is an important data point and could be used to support their hypothesis that clinic goers with TB symptoms are a high yield population to test.

-Only 44% of clinic goers underwent exit interview. I wonder if it is possible to also reconstruct the cascade of testing and testing results utilizing the registers from the clinic over the same time period. Although these registers may not include the results of the TB symptom screen, they would provide an objective snapshot of number of clinic goers offered testing, undergoing testing, and testing results and would be very informative.

-Additionally, it appears clinic attendees with TB symptoms were less likely to undergo exit interview. The authors should clarify how this may have biased their results. Perhaps participants who had TB symptoms did under HIV testing, but they weren’t captured for some reason in the exit interview?

-Is the reason for clinic appt recorded? If so, would also include in the models of testing. Perhaps certain presenting complaints (or even primary diagnosis of the encounter) are related to testing offer, uptake, and diagnosis.

Introduction/Methods

-Suggest clearly stating what the Malawian guidelines are for testing adults in a primary care clinic are.

-Authors mention these visits are for acute care, but my assumption it is for all-comers. Please clarify

Methods

-If possible, briefly describe what the pragmatic RCT will focus on.

-Describe what the Malawian PITC guidelines are.

-Is TB care provided at the clinic, or are patients referred somewhere else. What are the next steps for patients with TB symptoms at the clinic? Perhaps if they are referred somewhere else this could account for the fact that they were difficult to reach for exit interviews

-Be consistent RE: capitalization for Research Assistant.

-Clarify if PITC is to be offered for those with a negative test >90 days, or unknown HIV status (these details aren’t mention- I’m just assuming this is the guideline).

-Investigators mention excluding known PLHIV who were already taking ART, but what about known PLHIV not yet on ART? Were they also excluded? If so, clarify.

Results

-Are reasons for attendance to clinic, or clinic visit diagnosis available? Per above this may influence HIV testing as well.

-Was clinic flow different for patients who reported TB symptoms? Would they have been tested for HIV at another clinic (i.e. during further TB workup?).

-Were reasons test refusal among those offered testing recorded?

-Was time of last reported test recorded? Consider including in model.

Discussion

- Mentions that majority of health centers use symptom-based PITC, as opposed to universal offer. Is this in agreement with Malawi PITC guidelines?

-ROOT is mentioned for the first time in the Discussion. Suggest this be brought up earlier in the introduction to frame the reason to do the study.

-Discussion mentions WHO policy is in that setting with generalized epidemics that routine PITC should be offered to all with presumptive or diagnosed TB. Do the Malawian guidelines agree with this?

-The statistic that >50% of clinic attenders in the study reported TB symptoms is an important one and I’d suggest that this be mentioned in abstract.

-Authors mention that only 44% of participants underwent exit interview as a limitation. One way to overcome this limitation is to include data from registers from the study period regarding TB symptoms (if available), HIV testing offer, and HIV test result. The latter 2 should be in the register.

-Given the data, what do the authors suggest are the next steps?

References

-A minor comment, but please use consistent formatting/spacing in the references.

Table 1

Was any statistical testing done to see if there were differences between adult clinic attendees who had an exit interview or not. Suggest this is done. It appears that overall a lower proportion of participants who had a positive symptom screen underwent an exit interview.

Figure 1

For the top row of graphs please label Men and Women if that is what the Y axis is.

Reviewer #2: Authors have presented a nice study. They were interested to find out if adult patients seeking urgent care with TB-symptoms would receive PITC (as is recommended by WHO and Malawi National programmes).

Elegantly, they made their observations independent of the care offered at the health care facility (staffed by clinical officer, and nurses). Investigators positioned themselves at a registration desk for adult outpatients. Following verbal consent, identified patients through fingerprint biometrics and asked questions about age, HIV testing history, and TB symptoms.

Then, during exit interviews, 44% of adult patients were linked and patients were asked if an HIV test was offered, and if a test was taken.

Their data showed that contrary to their expectation, patients with TB symptoms were less likely to be tested than patients without TB patients. They also did statistical modeling to predict who would be more likely to be HIV tested.

The findings are clearly presented. This will be a useful contribution to the literature.

I have the following questions that should be addressed in the revision.

1. Participants were asked if they had ever previously tested for HIV. Can authors clarify if they collected the date of the last HIV test? If they did not, then mention this as a study limitation. If they did, they can provide the proportion who tested in the last year (annual testing for adults of the general population is a WHO recommendation).

2. Participants were asked their self-reported HIV status and ART status. HIV-positive patients on ART were excluded from the modeling. Can authors present the number of newly diagnosed patients (by self-report)? There should be patients who will have mentioned this, and thee majority will/should have started same day treatment. Unless authors plan to report on self-reported test outcome - but they should say something about self-reported HIV testing at care seeking (as they asked about it). Is it possible that patients with a new HIV diagnosis who were started on ART were not interested to present the exit desk for interview? This should be mentioned as a limitation of this study.

3. Qualitative data (e.g. focus group discussions with providers) were not collected. I find this another important limitation of the study. It would have been so easy to present result back to providers and ask them why they through that younger patients, men, and patients without TB symptoms were more like to be HIV tested.

Other points that can be clarified / improved upon:

1. Discussion hinges on two arguments - leading to two proposed strategies. Poor performance of PITC should be improved through targeted testing and ROOT (routine opt out testing).

Poor performance: Vast numbers of patients with TB-symptoms are not tested. These are major missed opportunities. Rather than calling this "suboptimal" - be more bold in your conclusion! This is very poor implementation of existing guidelines. Why is that so? that remains unclear. Authors suggest that targeted PITC could be achieved through oral self-testing.

ROOT is presented as a possible if not best strategy. Is this likely? Given that current system under performs?

Perhaps authors can more clearly distinguish the two strategies. Something like -- targeted PITC should be immediately implemented. ROOT is the preferred strategy but would require substantial resources.

Small points:

PITC HIV testing - misnomer - change to targeted HIV testing.

Is there a reference for ROOT?

Participants with a self-reported HIV testing history may drive them to request further HIV testing. Was this not explored in the exit interview? Were you offered an HIV test? Did you take an HIV test? or Did you ask for an HIV test yourself?

first paragraph discussion - last sentence. Optimism can be taken from an increase of 89.9% to 90.8%. There is either a typo in the stats, but i find this a minimal increase not worth discussing and no reason for optimism.

Last sentence of introduction (stratification by sex) reads as a methods sentence and is better placed in methods.

Approximately 1 in 4 patients is offered testing. And of those offered, 4 out of 5 accept HIV testing. Is it clear why a substantial number of patients refuse testing?

End.

6. PLOS authors have the option to publish the peer review history of their article (what does this mean?). If published, this will include your full peer review and any attached files.

Reviewer #1: No

Reviewer #2: Yes: Eduard Sanders

---

## [Author Response · Author response to Decision Letter 0]

23 Apr 2020

Dear Editors and Reviewers,

RE: Provider-initiated HIV testing and TB screening in the era of universal coverage: are the right people being reached? A cohort study in Blantyre, Malawi

I write on behalf of my co-authors to thank the reviewers for their careful review, and appreciate the opportunity to respond with a revised submission. Please see a point-by-point response below.

As recommended by the Reviewers, we have updated the manuscript to provide a more definite conclusion that the study shows poor implementation of national HIV testing guidelines and to emphasise that evaluation of additional strategies for increasing HIV testing rates are needed. The discussion has been expanded to further discuss potential strategies for improvement. This includes implementation of routine opt out HIV testing (ROOT), as well as the use of semi-supervised oral fluid-based HIV tests within facilities. 

We hope that with the changes made as suggested by the reviewers this manuscript can be published and provide motivation for a change in HIV testing policy.

Please find attached the requested documents:

- Manuscript

- Revised Manuscript with track changes

- Response To Reviewers

We have also included a copy of the patient questionnaire used during the trial as requested by the editors.

Yours sincerely,

Dr Luke Mair

---

## [Decision Letter · Decision Letter 1]

18 May 2020

PONE-D-19-35399R1

Provider-initiated HIV testing and TB screening in the era of universal coverage: are the right people being reached? A cohort study in Blantyre, Malawi

PLOS ONE

Dear Dr. Mair,

Thank you for submitting your manuscript to PLOS ONE. After careful consideration, we feel that it has improved in clarity but still but does not fully meet PLOS ONE’s publication criteria as it currently stands. Therefore, we invite you to submit a revised version of the manuscript that addresses the points raised during the review process.

Please see the specific suggestions from Reviewer 1 about additional clarification and some minor edits that would improve readability.

We would appreciate receiving your revised manuscript by Jul 02 2020 11:59PM. To enhance the reproducibility of your results, we recommend that if applicable you deposit your laboratory protocols in protocols.io, where a protocol can be assigned its own identifier (DOI) such that it can be cited independently in the future. For instructions see: http://journals.plos.org/plosone/s/submission-guidelines#loc-laboratory-protocols

We look forward to receiving your revised manuscript.

Kind regards,

Susan Marie Graham, MD, MPH, PhD

Academic Editor

PLOS ONE

Reviewers' comments:

Reviewer's Responses to Questions

**Comments to the Author**

1. If the authors have adequately addressed your comments raised in a previous round of review and you feel that this manuscript is now acceptable for publication, you may indicate that here to bypass the “Comments to the Author” section, enter your conflict of interest statement in the “Confidential to Editor” section, and submit your "Accept" recommendation.

Reviewer #1: (No Response)

Reviewer #2: All comments have been addressed

2. Is the manuscript technically sound, and do the data support the conclusions?

Reviewer #1: Yes

Reviewer #2: Yes

3. Has the statistical analysis been performed appropriately and rigorously? 

Reviewer #1: Yes

Reviewer #2: Yes

4. Have the authors made all data underlying the findings in their manuscript fully available?

Reviewer #1: Yes

Reviewer #2: Yes

5. Is the manuscript presented in an intelligible fashion and written in standard English?

Reviewer #1: (No Response)

Reviewer #2: Yes

6. Review Comments to the Author

Reviewer #1: Overall much improved. Suggests/comments follow.

Abstract

Clarify that this was among clinic attendees with acute illness. Dates are typically spelled out, but leave this style issue to the editors. Could consider simplifying abstract with regards to the categories of symptoms- since participants with and without symptoms the proportion offered testing was the same, why not just say among participants with any TB symptoms (especially since there appeared to be no difference among each specific symptom).

Agree with the conclusion of the Abstract.

Introduction

-Perhaps mention that Malawi is not one of the 7 countries to meet this 90/90/90 target.

-Add reference for ROOT and oral testing

Methods

-Provide reference of the Malawi national HIV Programme guidelines.

-Be explicit that patients with known HIV on ART were excluded in the Methods.

-Research assistant should be lower case I believe.

-What does opportunistic manner mean?

-Please be consistent with references either before or after a period.

-Not sure why there is a page break after Data and reproducability and Results?

Results

-Provide % for the 276 attenders who were HIV+ on ART that were excluded.

-Strikingly low offer of HIV testing (27.6%). (comment only)

-Striking that participants with a previous history of TB weren’t offered HIV testing. (comment only).

-Among those offered but not tested, what was the reason?

Discussion

-Please be careful with punctuation (2 periods at end of firs sentence in Discussion)

-Seems important to point out in the first paragraph of the discussion that this low rate of testing is despite the national guideline recommendations.

-ROOT already defined in Introduction, consider using only abbreviation here.

-Were there test kit stockouts during the time? It’s mentioned as a potential reason, but no further discussion.

-Although oral testing is a fantastic idea- why do the authors think this would increase testing? What does semi-supervised mean in regards to oral testing?

References

-Still has inconsistent formatting with references (some with hanging indent, some not).

-I’m a bit surprised that one of the national guideline referenced is from 2012.

Table 1

-Please use consistent capitalization for row headers.

-What are the numbers separated by a comma? Are they IQR? Ranges? Please specific in the table.

Table 2

Define WAIC

Reviewer #2: authors may want to check the following added sentence in their discussion:

"Qualitative research has suggested that patients often wish to know their status due a

period of health and therefore may be motivated to test themselves"

Not clear what is meant by a period of health

7. PLOS authors have the option to publish the peer review history of their article (what does this mean?). If published, this will include your full peer review and any attached files.

Reviewer #1: No

Reviewer #2: Yes: Eduard Sanders

---

## [Author Response · Author response to Decision Letter 1]

25 Jun 2020

RESPONSE TO REVIEWER 1’s COMMENTS

Comment 1:

Abstract

Clarify that this was among clinic attendees with acute illness. Dates are typically spelled out, but leave this style issue to the editors. Could consider simplifying abstract with regards to the categories of symptoms- since participants with and without symptoms the proportion offered testing was the same, why not just say among participants with any TB symptoms (especially since there appeared to be no difference among each specific symptom).

Thank you to the reviewer for these comments. In response additional clarification has been added to the abstract that these were participants with acute illness.

“In a prospective cohort study in Bangwe primary care clinic, Blantyre, Malawi, all adults (18 years or older) presenting with an acute illness were screened for TB symptoms (cough, fever, night sweats, weight loss).”

As suggested by the reviewer we have removed the percentages for each individual condition to make the abstract more readable. We have included an additional line in the text to explain that there was no difference between each symptoms.

“The proportions offered testing were similar among participants with and without TB symptoms (any TB symptom: 29.0% vs. 25.7%). This was consistent for each individual symptom; cough, weight loss, fever and night sweats.”

Comment 2:

Agree with the conclusion of the Abstract.

Thank you for these positive comments.

Comment 3:

Introduction

Perhaps mention that Malawi is not one of the 7 countries to meet this 90/90/90 target.

The authors agree that providing additional clarity here will add clarity for the reader. Therefore an additional sentence is added.

“Malawi is among the large proportion of countries still striving to reach these targets.”

Comment 4:

Add reference for ROOT and oral testing

Thank you. We have included references for each as follows.

ROOT: Silvestri et al. A comparison of HIV detection rates using routine opt-out provider-initiated HIV testing and counseling versus a standard of care approach in a rural African setting. J Acquir Immune Defic Syndr. 2011

Oral testing: Choko et al. 2015. Uptake, Accuracy, Safety, and Linkage into Care over Two Years of Promoting Annual Self-Testing for HIV in Blantyre, Malawi: A Community-Based Prospective Study. PLoS Med.

Comment 5:

Methods

-Provide reference of the Malawi national HIV Programme guidelines.

Thank you. This reference has now been included.

“Reference 3: Ministry of Health of Malawi. Malawi HIV Testing Services Guidelines. Lilongwe; 2016”

Comment 6:

-Be explicit that patients with known HIV on ART were excluded in the Methods.

Thank you. We have clarified this in the Methods:

“Participants with known HIV-positive status who were already taking ART were excluded from the analysis.”

Comment 7:

-Research assistant should be lower case I believe.

We are in agreement that these should be described in lower case and this change has been made.

Comment 8:

-What does opportunistic manner mean?

We have provided further clarity to this methodology as below.

“Research assistants were also based at the two clinic outpatient department exits, where they identified adults leaving the department and invited them to participate in an exit interview. The layout of the clinic is such that all participants exiting the clinic would pass the research assistants. Though researchers tried to selected as many patients as possible, due to limitations in the number of research assistants not all participants completed exit interview.”

Comment 9:

-Please be consistent with references either before or after a period.

This has been addressed and references now always appear after periods. Thank you.

Comment 10:

-Not sure why there is a page break after Data and reproducability and Results?

This has been removed.

Comment 11:

Results

-Provide % for the 276 attenders who were HIV+ on ART that were excluded.

This has now been included in the manuscript.

“A total of 276 (11.5%) clinic attenders reported that they were HIV positive and taking ART and were exclude from regression modelling.”

Comment 12:

-Strikingly low offer of HIV testing (27.6%). (comment only)

-Striking that participants with a previous history of TB weren’t offered HIV testing. (comment only).

Thank you for these comments. These show the suboptimal rates of testing offer in high risk populations which the manuscript aims to highlight.

Comment 13:

-Among those offered but not tested, what was the reason?

Rationale for participant’s non-completion of HIV testing was not evaluated in this study. A proportion of these patients may not have refused testing but instead were unable to undergo testing due to limited numbers of health-care workers or test kits. 

“These stated limitations may account for the population of patients in our study who were offered HIV testing but did not undergo a test.”

Comment 14:

Discussion

-Please be careful with punctuation (2 periods at end of firs sentence in Discussion)

This has been addressed.

Comment 15:

-Seems important to point out in the first paragraph of the discussion that this low rate of testing is despite the national guideline recommendations.

Thank you for this suggestion. A sentence has been added to the first paragraph of the discussion to highlight this key issue

“This highlights lack of implementation of national guidelines, which state that all patients attending an outpatient appointment should be offered an HIV test.”

Comment 16:

-ROOT already defined in Introduction, consider using only abbreviation here.

Corrected as recommended 

Comment 17:

-Were there test kit stockouts during the time? It’s mentioned as a potential reason, but no further discussion.

Thank you for this suggestion. No, there were no stock-outs during the study period.

Comment 18:

-Although oral testing is a fantastic idea- why do the authors think this would increase testing? What does semi-supervised mean in regards to oral testing?

A large body of evidence is now supporting HIV self-testing as a complementary strategy to increase uptake of HIV testing. Semi-supervised means that a health worker is in attendance, or available, should the patient require additional help in performing or interpreting the test themselves. 

Comment 19:

References

-Still has inconsistent formatting with references (some with hanging indent, some not).

Thank you. This has been addressed.

Comment 20:

-I’m a bit surprised that one of the national guideline referenced is from 2012.

Thank you. These are the most recent TB screening guidelines. WHO is currently updating its TB screening guidelines, and we anticipate updated Malawi guidelines thereafter.

Comment 21:

Table 1

-Please use consistent capitalization for row headers.

This has been addressed

Comment 22:

-What are the numbers separated by a comma? Are they IQR? Ranges? Please specific in the table.

Thank you. These are ranges (min and max), as indicated in the table row legend.

Comment 23:

Table 2

Define WAIC

Thank you. WAIC is the Watanabe-Akaike information criteria. This is a standard information criteria for assessing model fit.

RESPONSE TO REVIEWER TWO’S COMMENTS:

Comment 1

Authors may want to check the following added sentence in their discussion:

"Qualitative research has suggested that patients often wish to know their status due a period of health and therefore may be motivated to test themselves"

Not clear what is meant by a period of health

This sentence has been changed and now reads.

“Qualitative research has suggested that patients who noticed a decline in their health, may wish to know their status and therefore may be motivated to test themselves.”

---

## [Editor Report · Decision Letter 2]

8 Jul 2020

Provider-initiated HIV testing and TB screening in the era of universal coverage: are the right people being reached? A cohort study in Blantyre, Malawi

PONE-D-19-35399R2

Dear Dr. Mair,

We’re pleased to inform you that your manuscript has been judged scientifically suitable for publication and will be formally accepted for publication once it meets all outstanding technical requirements.

Kind regards,

Susan Marie Graham, MD, MPH, PhD

Academic Editor

PLOS ONE
---

## [Editor Report · Acceptance letter]

16 Jul 2020

PONE-D-19-35399R2 

Provider-initiated HIV testing and TB screening in the era of universal coverage: are the right people being reached? A cohort study in Blantyre, Malawi 

Dear Dr. Mair:

I'm pleased to inform you that your manuscript has been deemed suitable for publication in PLOS ONE. Congratulations! Your manuscript is now with our production department. 

Kind regards, 

on behalf of

Dr. Susan Marie Graham 

Academic Editor

PLOS ONE